# Accelerated Quality-Diversity through Massive Parallelism

**Bryan Lim**                                                     *bryan.lim16@imperial.ac.uk*
*Department of Computing*
*Imperial College London*

**Maxime Allard**                                                 *m.allard20@imperial.ac.uk*
*Department of Computing*
*Imperial College London*

**Luca Grillotti**                                                *luca.grillotti16@imperial.ac.uk*
*Department of Computing*
*Imperial College London*

**Antoine Cully**                                                 *a.cully@imperial.ac.uk*
*Department of Computing*
*Imperial College London*

**Reviewed on OpenReview:** *`https://openreview.net/forum?id=znNITCJyTI`*

## Abstract

Quality-Diversity (QD) optimization algorithms are a well-known approach to generate large collections of diverse and high-quality solutions. However, derived from evolutionary computation, QD algorithms are population-based methods which are known to be data-inefficient and require large amounts of computational resources. This makes QD algorithms slow when used in applications where solution evaluations are computationally costly. A common approach to speed up QD algorithms is to evaluate solutions in parallel, for instance by using physical simulators in robotics. Yet, this approach is limited to several dozen of parallel evaluations as most physics simulators can only be parallelized more with a greater number of CPUs. With recent advances in simulators that run on accelerators, thousands of evaluations can now be performed in parallel on single GPU/TPU. In this paper, we present QDax, an accelerated implementation of MAP-Elites which leverages massive parallelism on accelerators to make QD algorithms more accessible. We show that QD algorithms are ideal candidates to take advantage of progress in hardware acceleration. We demonstrate that QD algorithms can scale with massive parallelism to be run at interactive timescales without any significant effect on the performance. Results across standard optimization functions and four neuroevolution benchmark environments show that experiment runtimes are reduced by two factors of magnitudes, turning days of computation into minutes. More surprising, we observe that reducing the number of generations by two orders of magnitude, and thus having significantly shorter lineage does not impact the performance of QD algorithms. These results show that QD can now benefit from hardware acceleration, which contributed significantly to the bloom of deep learning.

## 1    Introduction

Quality-Diversity (QD) algorithms  (Pugh et al., 2016; Cully & Demiris, 2017; Chatzilygeroudis et al., 2021) have recently shown to be an increasingly useful tool across a wide variety of fields such as robotics (Cully et al., 2015; Chatzilygeroudis et al., 2018), reinforcement learning (RL) (Ecoffet et al., 2021), engineering design optimization (Gaier et al., 2018), latent space exploration for image generation (Fontaine & Nikolaidis, 2021) and video game design (Gravina et al., 2019; Fontaine et al., 2020a; Earle et al., 2021). Instead of

optimizing for a single solution like in conventional optimization, QD optimization searches for a population of high-performing and diverse solutions.

Adopting the QD optimization framework has many benefits. The diversity of solutions found in QD enables rapid adaptation and robustness to unknown environments (Cully et al., 2015; Chatzilygeroudis et al., 2018; Kaushik et al., 2020). Additionally, QD algorithms are also powerful exploration algorithms. They have been shown to be effective in solving sparse-reward hard-exploration tasks and achieved state-of-the-art results on previously unsolved RL benchmarks (Ecoffet et al., 2021). This is a result of the diversity of solutions present acting as stepping stones (Clune, 2019) during the optimization process. QD algorithms are also useful in the design of more open-ended algorithms (Stanley et al., 2017; Stanley, 2019; Clune, 2019) which endlessly generate their own novel learning opporunities. They have been used in pioneering work for open-ended RL with environment generation (Wang et al., 2019; 2020). Lastly, QD can also be used as effective data generators for RL tasks. The motivation for this arises from the availability of large amounts of data which resulted in the success of modern machine learning. The early breakthroughs in supervised learning in computer vision have come from the availability of large diverse labelled datasets (Deng et al., 2009; Barbu et al., 2019). The more recent successes in unsupervised learning and pre-training of large models have similarly come from methods that can leverage even larger and more diverse datasets that are unlabelled and can be more easily obtained by scraping the web (Devlin et al., 2018; Brown et al., 2020). As Gu et al. (Gu et al., 2021) highlighted, more efficient data generation strategies and algorithms are needed to obtain similar successes in RL.

Addressing the computational scalability of QD algorithms (focus of this work) offers a hopeful path towards open-ended algorithms that endlessly generates its own challenges and solutions to these challenges. Likewise, they can also play a significant role in this more data-centric view of RL by generating diverse and high-quality datasets of policies and trajectories both with supervision and in an unsupervised setting (Cully, 2019; Paolo et al., 2020).

The main bottleneck faced by QD algorithms are the large number of evaluations required that is on the order of millions. When using QD in the field of Reinforcement Learning (RL) for robotics, this issue is mitigated by performing these evaluations in physical simulators such as Bullet (Coumans & Bai, 2016–2020), DART (Lee et al., 2018), and MuJoCo (Todorov et al., 2012). However, these simulators have mainly been developed to run on CPUs. Methods like MPI can be used to parallelise over multiple machines, but this requires a more sophisticated infrastructure (i.e., multiple machines) and

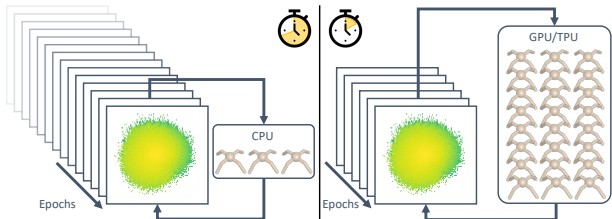

Figure 1: QDax uses massive parallelism on hardware accelerators like GPUs/TPUs to speed up runtime of QD algorithms by orders of magnitude

adds some network communication overhead which can add significant runtime to the algorithm. Additionally, the number of simulations that can be performed in parallel can only scale with the number of CPU cores available. Hence, both the lack of scalability coupled with the large number of evaluations required generally make the evaluation process of evolutionary based algorithms like QD, for robotics take days on modern 32-core CPUs. Our work builds on the advances and availability of hardware accelerators, high-performance programming frameworks (Bradbury et al., 2018) and simulators (Freeman et al., 2021; Makoviychuk et al., 2021) that support these devices to scale QD algorithms.

Historically, significant breakthroughs in algorithms have come from major advances in computing hardware. Most notably, the use of Graphic Processing Units (GPUs) to perform large vector and matrix computations enabled significant order-of-magnitude speedups in training deep neural networks. This brought about modern deep-learning systems which have revolutionized computer vision (Krizhevsky et al., 2012; He et al., 2016; Redmon et al., 2016), natural language processing (Hochreiter & Schmidhuber, 1997; Vaswani et al., 2017) and even biology (Jumper et al., 2021). Even in the current era of deep learning, significant network architectural breakthroughs (Vaswani et al., 2017) were made possible and are shown to scale with larger datasets and more computation (Devlin et al., 2018; Shoeybi et al., 2019; Brown et al., 2020; Smith et al., 2022).

Our goal in this paper is to bring the benefits of advances in compute and hardware acceleration to QD algorithms (Fig. 1). The key contributions of this work are: (1) We show that massive parallelization with QD using large batch sizes significantly speeds up run-time of QD algorithms at no loss in final performance, turning hours/days of computation into minutes. (2) We demonstrate that contrary to prior beliefs, the number of iterations (generations) of the QD algorithm is not critical, given a large batch size. We observe this across optimization tasks and a range of QD-RL tasks. (3) We release QDax, an open-source accelerated Python framework for Quality-Diversity algorithms (MAP-Elites) which enables massive parallelization on a single machine. This makes QD algorithms more accessible to a wider range of practitioners and researchers. The source code of QDax is available at `https://github.com/adaptive-intelligent-robotics/QDax`.

## 2  Related Work

**Quality-Diversity**. QD algorithms were derived from interests in divergent search methods (Lehman & Stanley, 2011a) and behavioural diversity (Mouret & Doncieux, 2009; 2012) in evolutionary algorithms and hybridizing such methods with the notion of fitness and reward (Lehman & Stanley, 2011b). While QD algorithms are promising solutions to robotics and RL, they remain computationally expensive and take a long time to converge due to high sample-complexity. The move towards more complex environments with high-dimensional state and action spaces, coupled with the millions of evaluations required for the algorithm to converge, makes these algorithms even more inaccessible to regular hardware devices. Progress has been made towards lowering the sample complexity of QD algorithms and can generally be categorized into two separate approaches. The first approach is to leverage the efficiency of other optimization methods such as evolution strategies (Colas et al., 2020; Fontaine et al., 2020b; Cully, 2020; Wang et al., 2021) and policy-gradients (Nilsson & Cully, 2021; Pierrot et al., 2021). The other line of work known as *model-based quality-diversity* (Gaier et al., 2018; Keller et al., 2020; Lim et al., 2021), reduces the number of evaluations required through the use of surrogate models to provide a prediction of the descriptor and objective.

Our work takes a separate approach orthogonal to sample-efficiency and focuses improvement on the runtime of QD algorithms instead by leveraging the batch size at each iteration. Additionally, despite algorithmic innovations that improve sample-efficiency, most QD implementations still rely on evaluations being distributed over large compute systems. These often give impressive results (Colas et al., 2020; Fontaine et al., 2019) but such resources are mainly inaccessible to most researchers and yet still take significant amount of time to obtain. Our work aims to make QD algorithms more accessible by running quickly on more commonly available accelerators, such as cloud available GPUs.

**Hardware Acceleration for Machine Learning.** Machine Learning, and more specifically Deep Learning methods, have benefited from specialised hardware accelerators that can parallelize operations. In the mid-2000's researchers started using GPUs to train neural networks (Steinkrau et al., 2005) because of their high degree of parallelism and high memory bandwidth. After the introduction of general purpose GPUs, the use of specialized GPU compatible code for Deep Learning methods (Raina et al., 2009; Ciresan et al., 2012) enabled deep neural networks to be trained a few orders of magnitude quicker than previously on CPUs (Lecun et al., 2015). Very quickly, frameworks such as Torch (Collobert et al., 2011), Tensorflow (Abadi et al., 2016), PyTorch (Paszke et al., 2019) or more recently JAX (Bradbury et al., 2018) were developed to run numerical computations on GPUs or other specialized hardware.

These frameworks have led to tremendous progress in deep learning. In other sub-fields such as deep reinforcement learning (DRL) or robotics, the parallelization has happened on the level of neural networks. However, RL algorithms need a lot of data and require interaction with the environment to obtain it. Such methods suffer from a slow data collection process as the physical simulators used to collect data were mainly developed for CPUs, which results in a lack of scalable parallelism and data transfer overhead between devices. More recently, new rigid body physics simulators that can leverage GPUs and run thousands of simulations in parallel have been developed. Brax (Freeman et al., 2021) and IsaacGym (Makoviychuk et al., 2021) are examples of these new types simulators. Gradient-based DRL methods can benefit from this massive parallelism as this would directly correspond to estimating the gradients more accurately through collection of larger amounts of data in a faster amount of time (Rudin et al., 2021) at each optimization step. Recent work (Gu et al., 2021; Rudin et al., 2021) show that control policies can be trained with DRL algorithms like

PPO (Schulman et al., 2017) and DIAYN (Eysenbach et al., 2018) in minutes on a single GPU. However, unlike gradient-based methods, it is unclear until now how evolutionary and population-based approaches like QD would benefit from this massive parallelism since the implications of massive batch sizes has not been studied to the best of our knowledge. Our work studies the impact of massive parallelization on QD algorithms and its limitations.

## 3 Problem Statement

**Quality-Diversity Problem.** The Quality-Diversity (QD) problem Pugh et al. (2016); Chatzilygeroudis et al. (2021); Fontaine & Nikolaidis (2021) is an optimization problem which consists of searching for a set of solutions $\mathcal{A}$ that (1) are locally optimal, and (2) exhibit diverse features. QD problems are characterized by two components: (1) an objective function to maximize $f : \Theta \to \mathbb{R}$, and (2) a descriptor function $d : \Theta \to \mathcal{D} \subseteq \mathbb{R}^n$. That descriptor function is used to differentiate between solutions; it takes as input a solution $\boldsymbol{\theta} \in \Theta$, and computes a descriptive low-dimensional feature vector.

The goal of QD algorithms is to return an *archive* of solutions $\mathcal{A}$ satisfying the following condition: for each achievable descriptor $\boldsymbol{c} \in \mathcal{D}$, there exists a solution $\boldsymbol{\theta}_{\mathcal{A},\boldsymbol{c}} \in \mathcal{A}$ such that $d(\boldsymbol{\theta}_{\mathcal{A},\boldsymbol{c}}) = \boldsymbol{c}$ and $f(\boldsymbol{\theta}_{\mathcal{A},\boldsymbol{c}})$ maximizes the objective function with the same descriptor $\{f(\boldsymbol{\theta}) \mid \boldsymbol{\theta} \in \Theta \wedge d(\boldsymbol{\theta}) = \boldsymbol{c}\}$. However, the descriptor space $\mathcal{D}$ is usually continuous. This would require storing an infinite amount of solutions in the set. QD algorithms alleviate this problem by considering a tesselation of the descriptor space into cells (Mouret & Clune, 2015; Cully et al., 2015; Vassiliades et al., 2017): $(\text{cell}_i)_{i \in \mathcal{I}}$ and keeping only a single solution per cell. QD algorithms aim at finding a set of policy parameters $(\boldsymbol{\theta}_j)_{j \in \mathcal{J}}$ maximizing the QD-Score Pugh et al. (2016), defined as follows (where $f(\cdot)$ is assumed non-negative without loss of generality):

$$\underset{(\boldsymbol{\theta}_j)_{j \in \mathcal{J}}}{\text{maximize}} \quad \text{QD-Score} = \sum_{j \in \mathcal{J}} f(\boldsymbol{\theta}_j) \quad \text{such that } \forall j \in \mathcal{J}, \, d(\boldsymbol{\theta}_j) \in \text{cell}_j \tag{1}$$

Thus, maximizing the QD-Score is equivalent to maximizing the number of cells containing a policy from the archive, while also maximizing the objective function in each cell.

**Quality-Diversity for Neuroevolution-based RL.** QD algorithms can also be used on RL problems, modeled as Markov Decision Processes $(\mathcal{S}, \mathcal{A}, p, r)$, where $\mathcal{S}$ is the states set, $\mathcal{A}$ denotes the set of actions, $p$ is a probability transition function, and $r$ is a state-dependent reward function. The return $R$ is defined as the sum of rewards: $R = \sum_t r_t$. In a standard RL setting, the goal is to find a policy $\pi_{\boldsymbol{\theta}}$ maximizing the expected return.

The QD for Reinforcement Learning (QD-RL) problem (Nilsson & Cully, 2021; Tjanaka et al., 2022) is defined as a QD problem in which the goal is to find a set of diverse policy parameters $\mathcal{A} = (\boldsymbol{\theta}_j)_{j \in \mathcal{J}}$ leading to diverse high-performing behaviours (Fig. 2). In the QD-RL context, the objective function matches the expected return $f(\boldsymbol{\theta}_j) = \mathrm{E}\left[R^{(\boldsymbol{\theta}_j)}\right]$; the descriptor function $d(\cdot)$ characterizes the state-trajectory of policy $\pi_{\boldsymbol{\theta}_j}$: $d(\boldsymbol{\theta}_j) = \mathrm{E}\left[\widetilde{d}(\boldsymbol{\tau}^{(\boldsymbol{\theta}_j)})\right]$ (with $\boldsymbol{\tau}$ denoting the state-trajectory $\boldsymbol{s}_{1:T}$). The QD-Score to maximize (formula 1) can then be expressed as follows, where all expected returns should be non-negative:

$$\underset{(\boldsymbol{\theta}_j)_{j \in \mathcal{J}}}{\text{maximize}} \quad \text{QD-Score} = \sum_{j \in \mathcal{J}} \mathrm{E}\left[R^{(\boldsymbol{\theta}_j)}\right] \quad \text{such that } \forall j \in \mathcal{J}, \, \mathrm{E}\left[\widetilde{d}(\boldsymbol{\tau}^{(\boldsymbol{\theta}_j)})\right] \in \text{cell}_j \tag{2}$$

## 4 Background: MAP-Elites

MAP-Elites Mouret & Clune (2015) is a well-known QD algorithm which considers a descriptor space discretized into grid cells (Fig. 3 4th column). At the start, an archive $\mathcal{A}$ is created and initialized by evaluating random solutions. Then, at every subsequent iteration, MAP-Elites (i) generates new candidate solutions, (ii) evaluates their return and descriptor and (iii) attempts to add them to the archive $\mathcal{A}$. The iterations are done until we reach a total budget $H$ of evaluations. During step (i), solutions are selected uniformly from the archive $\mathcal{A}$, and undergo variations to obtain a new batch of solutions $\widetilde{\mathcal{B}}$. In all our experiments, we use the iso-line variation operator (Vassiliades & Mouret, 2018) (Appendix Algo. 2).

Then (ii), the solutions in the sampled batch $\widetilde{\mathcal{B}} = (\widetilde{\boldsymbol{\theta}}_j)_{j \in [\![1, N_{\mathcal{B}}]\!]}$ are evaluated to obtain their respective returns $(R^{(\widetilde{\boldsymbol{\theta}}_j)})_{j \in [\![1, N_{\mathcal{B}}]\!]}$ and descriptors $(d(\widetilde{\boldsymbol{\theta}}_j))_{j \in [\![1, N_{\mathcal{B}}]\!]}$. Finally (iii), each solution $\widetilde{\boldsymbol{\theta}}_j \in \mathcal{A}$ is placed in its corresponding cell in the behavioural grid according to its descriptor $d(\widetilde{\boldsymbol{\theta}}_j)$. If the cell is empty, the solution is added to the archive. If the cell is already occupied by another solution, the solution with the highest return is kept, while the other is discarded. A pseudo-code for MAP-Elites is presented in Algo. 1.

We use MAP-Elites to study and show how QD algorithms can be scaled through parallelization. We leave other variants and enhancements of QD algorithms which use learned descriptors (Cully, 2019; Paolo et al., 2020; Miao et al., 2022) and different optimization strategies such as policy gradients (Nilsson & Cully, 2021; Pierrot et al., 2021; Tjanaka et al., 2022) and evolutionary strategies (Colas et al., 2020; Fontaine et al., 2020b) for future work; we expect these variants to only improve performance on tasks, and benefit from the same contributions and insights of this work.

## 5 Leveraging Hardware Acceleration for Quality-Diversity

In population-based methods, new solutions are the result of older solutions that have undergone variations throughout an iterative process as described in Algo. 1. The number of iterations $I$ of a method commonly depends on the total number of evaluations (i.e. computational budget) $H$ for the optimization algorithm and the batch size $N_{\mathcal{B}}$. For a fixed computation budget, a large batch size $N_{\mathcal{B}}$ would result in a lower number of iterations $I$ and vice versa. At each iteration, a single solution can undergo a variation and each variation is a learning step to converge towards an optimal solution. It follows that the number of iterations $I$ defines the maximum number of learning steps a solution can take. In the extreme case of $N_{\mathcal{B}} = H$, the method simply reduces to a single random variation to a random sample of parameters.

---

**Algorithm 1** MAP-Elites ($N_{\mathcal{B}}$: Batch size)

---

1: **for** iteration $\in [\![1, I]\!]$ **do**
2:      **if** first iteration **then**
3:          $\mathcal{B} \leftarrow$ random solutions
4:      **else**
5:          $\mathcal{B} \leftarrow$ select solutions from archive $\mathcal{A}$
6:      $\widetilde{\mathcal{B}} = (\widetilde{\boldsymbol{\theta}}_j)_{j \in [\![1, N_{\mathcal{B}}]\!]} \leftarrow \text{VARIATION}(\mathcal{B})$[a]
7:      **for** $j \in [\![1, N_{\mathcal{B}}]\!]$ **do**
8:          run episode of $\pi_{\widetilde{\boldsymbol{\theta}}_j}$, get return $R^{(\boldsymbol{\theta}_j)}$ and traj. $\boldsymbol{\tau}^{(\boldsymbol{\theta}_j)}$
9:          cell $\leftarrow$ get grid cell of descriptor $\widetilde{d}(\boldsymbol{\tau}^{(\boldsymbol{\theta}_j)})$
10:         $\boldsymbol{\theta}_{\text{cell}} \leftarrow$ get content of cell
11:         **if** $\boldsymbol{\theta}_{\text{cell}}$ is None **then**
12:             Add $\boldsymbol{\theta}_j$ to cell
13:         **else if** $R^{(\boldsymbol{\theta}_j)} > R^{(\boldsymbol{\theta}_{\text{cell}})}$ **then**
14:             Replace $\boldsymbol{\theta}_{\text{cell}}$ with $\boldsymbol{\theta}_j$ in cell
15:         **else**
16:             Discard $\boldsymbol{\theta}_j$
     **return** archive $\mathcal{A}$

---

[a]Any variation operator can be used to obtain new solutions. We use the iso-line variation (see Algo 2 in Appendix. A.2)

Based on this, an initial thought would be that the performance of population-based methods would be negatively impacted by a heavily reduced number of learning steps and require more iterations to find good performing solutions. The iterations of the algorithm are a sequential operation and cannot be parallelized. However, the evaluation of solutions at each iteration can be massively parallelized by increasing the batch size $N_{\mathcal{B}}$ which suits modern hardware accelerators. We investigate the effect of large $N_{\mathcal{B}}$ on population-based methods (more specifically QD algorithms) by ablating exponentially increasing $N_{\mathcal{B}}$ which was relatively unexplored in the literature.

Conventional QD algorithms parallelize evaluations by utilizing multiple CPU cores, where each CPU separately runs an instance of the simulation to evaluate a solution. For robotics experiments, we utilize Brax (Freeman et al., 2021), a differentiable physics engine in Python which enables massively parallel rigid body simulations. By leveraging a GPU/TPU, utilizing this simulator allows us to massively parallelize the evaluations in the QD loop which is the major bottleneck of QD algorithms. To provide a sense of scale, QD algorithms normally run on the order of several dozens of evaluations in parallel with $N_{\mathcal{B}} \sim 10^2$ due to limited CPUs while Brax can simulate over 10,000 solutions in parallel allowing QDax to have $N_{\mathcal{B}} \sim 10^5$. Brax is built on top of the JAX (Bradbury et al., 2018) programming framework, which provides an API to run accelerated code across any number of hardware acceleration devices such as CPU, GPU or TPU.

Beyond massive parallelization, acceleration of our implementation is also enabled with code compatible with just-in-time (JIT) compilation and fully on-device computation. Another key benefit of JAX is the JIT compilation which allows JAX to make full use of its Accelerated Linear Algebra (XLA).We provide

implementation details with static archives that makes QDax compatible with JIT in the Appendix. Lastly, another bottleneck which slowed the algorithm down was the data transfer and marshalling across devices. To address this issue, we carefully consider data structures and place all of them on-device. QDax places the JIT-compiled QD algorithm components on the same device. This enables the entire QD algorithm to be run without interaction with the CPU.

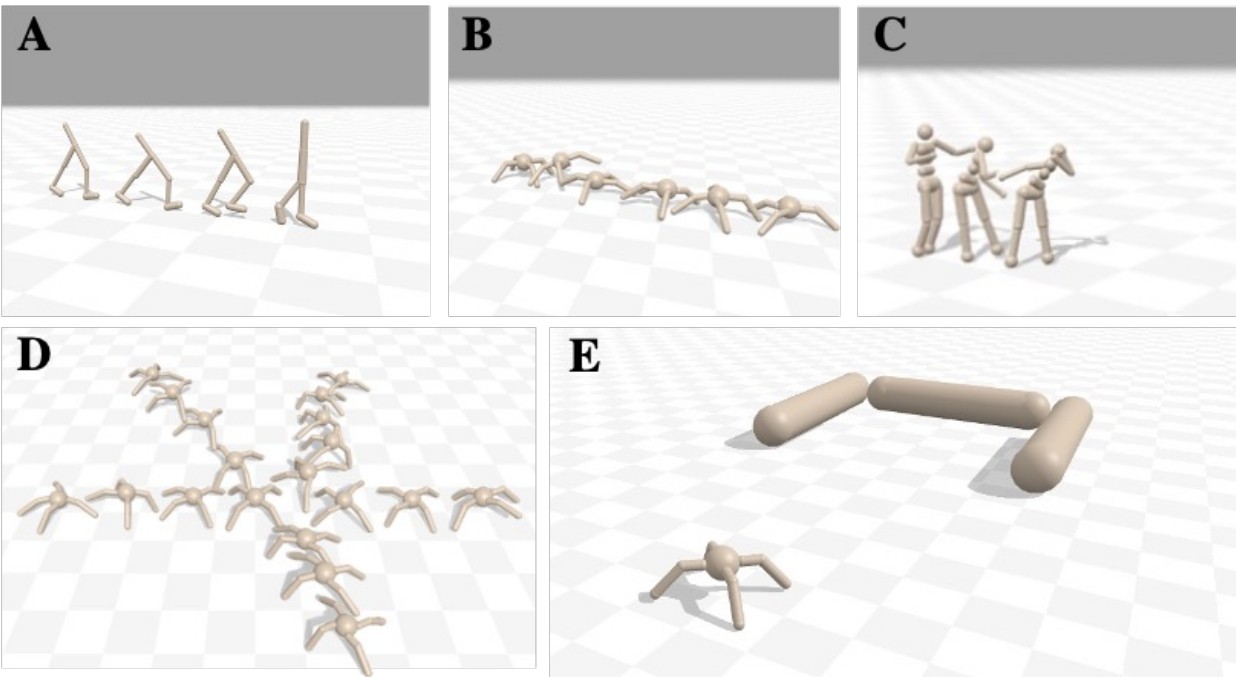

Figure 2: QD algorithms can discover diverse solutions. A, B,C: Uni-directional Walker, Ant and Humanoid task respectively discover diverse gaits for moving forward, D: Omni-directional Ant task discovers ways to move in every direction, E: Ant Trap task with deceptive rewards.

## 6 Experiments

Our experiments aim to answer the following questions: (1) How does massive parallelization affect performance of Quality-Diversity (MAP-Elites) algorithms? (2) How significant is the number of iterations/learning steps in QD algorithms? (3) What magnitude of speed-up does massive parallelization offer over existing implementations? (4) How does this differ across different hardware accelerators?

### 6.1 Domains

**Rastrigin and Sphere.** Rastrigin and Sphere are standard optimization functions commonly used as benchmark domains in optimization (Hansen et al., 2010; 2021) and QD-literature (Fontaine et al., 2020b; Fontaine & Nikolaidis, 2021). We optimize a $n = 100$ dimensional parameter space bounded between 0 and 1. More details regarding the objective function and descriptors are provided in the Appendix. We select these simple domains to demonstrate that the phenomena of large batch sizes applies generally to QD algorithms and not just to certain domains.

**Planar Arm.** The planar arm is a simple low-dimensional control task used in QD literature (Cully et al., 2015; Vassiliades & Mouret, 2018; Fontaine & Nikolaidis, 2021) where the goal is to find the inverse kinematic solution (joint positions) of a planar robotic arm for every reachable position of the end effector. As the arm is redundant, the objective $f$ for each solution is to minimize the variance of the joint angles (i.e. smooth solutions) while the descriptor corresponds to the $x$-$y$ position of the end effector obtained via forward kinematics. For our experiments, we use a 7-DoF arm.

**Continuous Control QD-RL.** We perform experiments on three different QD-RL benchmark tasks (Cully et al., 2015; Conti et al., 2018; Nilsson & Cully, 2021; Tjanaka et al., 2022); *omni-directional* robot locomotion, *uni-directional* robot locomotion and a deceptive reward *trap* task. In the omni-directional task, the goal is to discover locomotion skills to move efficiently in every direction.The descriptor functions is defined as the final $x$-$y$ positions of the center of mass of the robot at the end of the episode while the objective $f$ is defined as a sum of a survival reward and torque cost. In contrast, the goal in the uni-directional tasks is to find a collection of diverse gaits to walk forward as fast as possible. In this task, the descriptor function is defined as the average time over the entire episode that each leg is in contact with the ground. For each foot $i$, the contact with the ground $C_i$ is logged as a Boolean (1: contact, 0: no-contact) at each time step $t$. The descriptor function of this task was used to allow robots to recover quickly from mechanical damage (Cully et al., 2015). The objective $f$ of this task is a sum of the forward velocity, survival reward and torque cost. Full equation details can be found in the Appendix B. In the Trap tasks, the environment contains a trap 2 right in front of the ant. The goal is to learn to move forward as fast as possible. If naively done using purely objective based algorithms, the ant will get stuck in the trap. The objective $f$ of this task is a sum of the forward velocity, survival reward and torque cost while the descriptor is the final $x$-$y$ position of the robot at the end of the episode.

We use the Hopper, Walker2D, Ant and Humanoid gym locomotion environments made available on Brax (Freeman et al., 2021) on these tasks. In total, we report results on a combination of six tasks and environments; Omni-directional Ant, Uni-directional Hopper, Walker, Ant and Humanoid, and Ant Trap. Fig. 2 illustrates examples of the types behaviors discovered from these tasks. We use fully connected neural network controllers with two hidden layers of size 64 and tanh output activation functions as policies across all QD-RL environments and tasks.

## 6.2 Effects of massive parallelization on QD algorithms

To evaluate the effect of the batchsize $N_{\mathcal{B}}$, we run the algorithm for a fixed number of evaluations. We use 5 million evaluations for all QD-RL environments and 20 million evaluations for *rastrigin* and *sphere*. We evaluate the performance of the different batch sizes using the QD-score. The QD-score (Pugh et al., 2016) aims to capture both performance and diversity in a single metric. This metric is computed as the sum of objective values of all solutions in the archive (Eqn. 1). We plot this metric with respect to three separate factors: number of evaluations, number of iterations and total runtime. Other commonly used metrics in QD literature such as the best objective value and coverage can be found in the Appendix. We use a single A100 GPU to perform our experiments.

Fig. 3 shows the performance curves for QD-score and more importantly the differences when plot against the number of evaluations, iterations and total runtime. A key observation in the first column is that the metrics converge to the same final score after the fixed number of evaluations regardless of the batch size used. The Wilcoxon Rank-Sum Test for the final QD score across all the different batches results in p-values p>0.05 after applying the Bonferroni Correction. This shows that we do not observe statistically significant differences between the different batch sizes. Therefore, larger batch sizes and massive parallelism do not negatively impact the final performance of the algorithm. However, an important observation is that the larger batch sizes have a trend to be slower in terms of number of evaluations to converge. This can be expected as a larger number of evaluations are performed per iteration at larger batch sizes. Conversely, in some cases (Ant-Uni), a larger batch size can even have a positive impact on the QD-score. Given this result, the third column of Fig. 3 then demonstrates the substantial speed-up in total runtime of the algorithm obtained from using larger batch sizes with massive parallelism while obtaining the same performances. We can obtain similar results but in the order of minutes instead of hours. An expected observation we see when comparing the plots in the second and third column of Fig. 3 is how the total runtime is proportional to the number of iterations. As we are able to increase the evaluation throughput at each iteration through the parallelization, it takes a similar amounts of time to evaluate both smaller and larger batch sizes. The speed-up in total runtime of the algorithm by increasing batch size eventually disappears as we reach the limitations of the hardware. This corresponds to the results presented in the the next section 6.3 (Fig. 4) where we see the number eval/s plateauing.

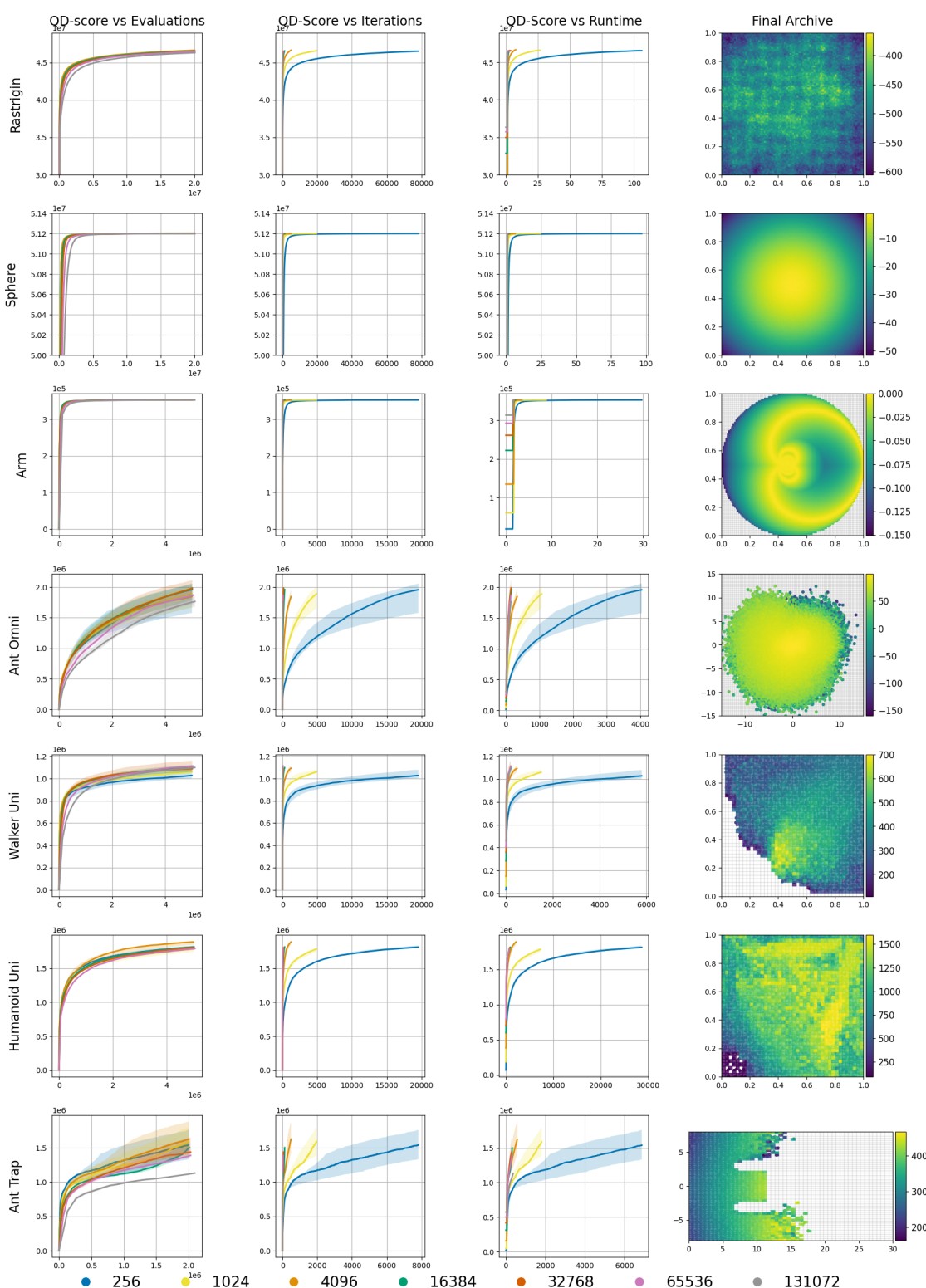

Figure 3: Performance metrics across domains. The plots show the QD-Score against the number of evaluations, iterations and run-times for each batch size. The rightmost column shows the final archives. The Ant and Hopper Uni results are presented in the Appendix. The bold lines and shaded areas represent the median and interquartile range over 10 replications respectively.

| Batch Size | Black-box | | Simple Control | Neuroevolution for QD-RL | | | | | |
| | Rastrigin | Sphere | Arm | HopperUni | WalkerUni | AntUni | HumanoidUni | AntOmni | AntTrap |
| --- | --- | --- | --- | --- | --- | --- | --- | --- | --- |
| 256 | 55025 | 65550 | 7761 | 185 | 8587 | 14951 | 11183 | 7272 | 1339 |
| 1,024 | 12720 | 16327 | 1770 | 63 | 1432 | 3412 | 3608 | 1690 | 562 |
| 4,096 | 2920 | 4086 | 451 | 15 | 254 | 734 | 470 | 400 | 119 |
| 16,384 | 832 | 1053 | 104 | 4 | 62 | 177 | 182 | 98 | 63 |
| 32,768 | 455 | 535 | 54 | 2 | 29 | 85 | 101 | 49 | 24 |
| 65,536 | 240 | 281 | 34 | 2 | 21 | 45 | 56 | 33 | 18 |
| 131,072 | 147 | 151 | 23 | 1 | 13 | 26 | - | 21 | 16 |

Table 1: Number of iterations needed to reach threshold QD-score (minimum QD-score across all batch sizes) for the range of batch sizes. The medians over the 10 replications are reported.

The most surprising result is that the number of iterations and thus learning steps of the algorithm do not significantly affect the performance of the algorithm when increasing batch sizes $N_{\mathcal{B}}$ are used (Fig. 3 - second column). In the case of the *QD-RL* domains, we observe that using $N_{\mathcal{B}} = 131,072$ which runs for a total of only $I = 39$ provides the similar performance as when run with $N_{\mathcal{B}} = 256$ and $I = 19,532$. This is true for all the problem domains presented. To evaluate this more concretely, Table 1 shows the iterations needed to reach a threshold QD-score. The threshold QD-score is the minimum QD-score reached across all the batch sizes. The results in Fig. 1 clearly show that larger batch sizes require significantly less iterations to achieve the same QD-score.

Given a fixed number of evaluations, a larger batch size would imply a lower number of iterations. This can also be observed in the second column of Fig. 3. Therefore, our results show that a larger batch size with less iterations has no negative impact on QD algorithms and can significantly speed-up the runtime of QD algorithms using parallelization for a fixed evaluation budget. The iterations/learning steps remain an important part of QD algorithms as new solutions that have been recently discovered and added to the archive $\mathcal{A}$ from a previous iteration can be selected as stepping stones to form good future solutions. This is particularly evident in more complex tasks (Ant Trap) where there is an exploration bottleneck in which it can be observed that large batch sizes struggle due to the lack of stepping stones from a low number of iterations. However, our experiments show that iterations are not as important as long as the number of evaluations remains identical. This is a critical observation as while iterations cannot be performed in parallel, evaluations can be, which enables the use of massive parallelization.

This is in contrast to other population-based approaches such as evolutionary strategies (Salimans et al., 2017), where we observe that massively large batch sizes can be detrimental when we run similar batch size ablation experiments (see Appendix C). We hypothesize that the reason for this is because in conventional evolutionary strategies (ES), there is a single optimization "thread" going on. The population is mainly used to provide an empirical approximation of the gradient. Increasing the size of the population helps to obtain a better estimate, but after a certain size, there are no benefits. On the contrary, with larger batch sizes, the algorithms will waste evaluations. Conversely, in Quality-Diversity algorithms (MAP-Elites in particular), each of the thousands cells of the container can be seen as an independent optimization "thread". This enables MAP-Elites to greatly benefit from extra-large batch sizes.

### 6.3 Evaluation throughput and runtime speed of QD algorithms

We also evaluate the effect that increasing batch sizes have on the evaluation throughput of QD algorithms. We start with a batch size $N_{\mathcal{B}}$ of 64 and double from this value until we reach a plateau and observe a drop in performance in throughput. In our experiments, a maximum batch size of 131,072 is used.

The number of evaluations per second (eval/s) is used to quantify this throughput. The eval/s metric is computed by running the algorithm for a fixed number of generations (also referred to as iterations) $N$ (100 in our experiments). We divide the corresponding batch size $N_{\mathcal{B}}$ representing the number of evaluations performed in this iteration and divide this value by the time it takes to perform one iteration $t_n$. We use the final average value of this metric across the entire run: $eval/s = \frac{1}{N} \sum_{n=1}^{N} \frac{N_{\mathcal{B}}}{t_n}$. While the evaluations per second can be an indication of the improvement in throughput from this implementation, we ultimately care about running the entire algorithm faster. To do this, we evaluate the ability to speed up the total runtime of QD algorithms. In this case, we run the algorithm to a fixed number of evaluations (1 million), as

| Implementation | Simulator | Resources | Eval/s | Batch size | Runtime (s) | Batch size |
|---|---|---|---|---|---|---|
| QDax (Ours) | Brax | GPU A100 | 30,846 | 65,536 | **69** | 65,536 |
| QDax (Ours) | Brax | GPU 2080 | 11,031 | 8,192 | 117 | 8,192 |
| pyribs | PyBullet | 32 CPU-cores | 184 | 8,192 | 7,234 | 4,096 |
| pymapelites | PyBullet | 32 CPU-cores | 185 | 8,192 | 6,509 | 16,384 |
| Sferes$_{v2}$ | DART | 32 CPU-cores | 1,190 | 512 | 1,243 | 32,768 |

Table 2: Maximum throughput of evaluations per second and fastest run time obtained and their corresponding batch sizes across implementations. The medians over the 10 replications are reported.

usually done in QD literature. Running for a fixed number of iterations would be an unfair comparison as the experiments with smaller batch sizes would have much less evaluations performed in total.

For this experiment, we consider the Ant Omnidirectional task. We compare against common implementations of MAP-Elites and open-source simulators which utilize parallelism across CPU threads (see Table 2). All baseline algorithms used simulations with a fixed number of timesteps (100 in our experiments). We compare against both Python and C++ implementations as baselines. Pymapelites (Mouret & Clune, 2015) is a simple reference implementation from the authors of MAP-Elites that was made to be easily transformed for individual purposes. Pyribs (Tjanaka et al., 2021) is a more recent QD optimization library maintained by the authors of CMA-ME (Fontaine et al., 2020b). In both Python implementations, evaluations are parallelised on each core using the multiprocessing Python package. Lastly,

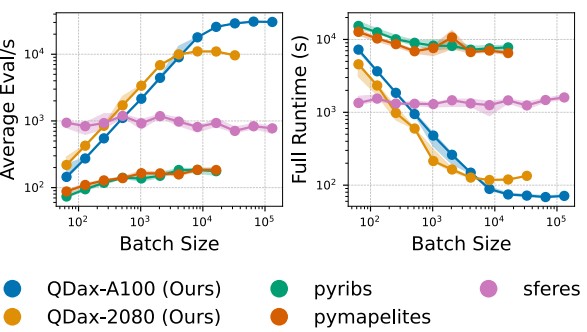

Figure 4: Average number of eval/s and full runtime of algorithm across batch sizes and implementations. Note the log scales on both axes to make distinction between batch sizes clearer.

Sferes$_{v2}$ (Mouret & Doncieux, 2010) is an optimized, multi-core and lightweight C++ framework for evolutionary computation, which includes QD implementations. It relies on template-based programming to achieve optimized execution speeds. The multi-core distribution for parallel evaluations is handled by Intel Threading Building Blocks (TBB) library. For simulators, we use PyBullet (Coumans & Bai, 2016–2020) in our Python baselines and Dynamic Animation and Robotics Toolkit (DART) (Lee et al., 2018) for the C++ baseline. We recognize that this comparison is not perfect and there could exist more optimized combinations of implementations and simulators but believe the baselines selected gives a good representation of what is commonly used in most works Cully & Mouret (2013); Nilsson & Cully (2021); Tjanaka et al. (2022).

We also test our implementation on two different GPU devices, a more accessible RTX2080 local device and a higher-performance A100 on Google Cloud. We only consider a single GPU device at each time. QDax was also tested and can be used to perform experiments across distributed GPU and TPU devices but we omit these results for simplicity.

Fig. 4 (Left) clearly shows that QDax has the ability to scale to much larger batch sizes which results in a higher throughput of evaluations. It is important to note the log scale on both axes to appreciate the magnitude of the differences. For QDax implementations (blue and orange), the number of evaluations per second scales as the batch size used increases. This value eventually plateaus once we reach the limit of the device. On the other hand, all the baseline implementations scales to a significantly lower extent. These results can be expected as evaluations using simulators which run on CPUs are limited as each CPU core runs a separate instance of the simulation. Therefore, given only a fixed number of CPU cores, these baselines would not scale as the batch size is increased. Scaling is only possible by increasing the number of CPUs used in parallel which is only possible with large distributed system with thousands of CPUs in the network. QDax can reach up to a maximum of 30,000 evaluations per second on an A100 GPU compared to maximum of 1,200 (C++) or 200 (Python) evaluations per second in the baselines (see Table 2). This is a 30 to 100 times increase in throughput, turning computation on the order of days to minutes. The negligible differences between the pyribs (green) and pymapelites (red) results show that the major bottleneck is indeed

the evaluations and simulator used, as both of these baselines use the PyBullet simulator. The performance of the Sferes$_{v2}$ (purple) implementation can be attributed to its highly optimized C++ code. However, the same lack of scalability is also observed when the batchsize is increased. When looking at run time of the algorithm for a fixed number of evaluations on Fig. 4 (Right), we can see the effect of the larger throughput of evaluations at each iteration reflected in the decreasing run-time when larger batch sizes are used. We can run a QD algorithm with 1 million evaluations in just slightly over a minute (See Table 2) when using a batch size of 65,536 compared to over 100 minutes taken by Python baselines.

Our results also show that this scaling through massive parallelism is only limited by the hardware available. The experiments on both the RTX2080 (orange) and A100 (blue) show similar trends and increases in both evaluations per second and total runtime. The 2080 plateaus at a batch size of 8,192 capable of  11,000 eval/s while the higher-end A100 plateaus later at a batch size of 65,536 completing  30,000 eval/s.

## 7  Limitations and Future Work

In this paper, we presented QDax, an implementation of MAP-Elites that utilizes massive parallelization on accelerators that reduce the runtime of QD algorithms to interactive timescales on the order of minutes instead of hours or days. We evaluate QDax across a range QD tasks and show that the performance of QD algorithms are maintained despite the significant speed-up that comes with the massive parallelism. Despite the apparent importance of iterations in QD algorithms, we show that when large batch sizes are used, a heavily reduced number of iterations and hence learning steps, provides similar results thus greatly accelerating the runtime of these QD algorihtms. This is observed on all the QD problems we considered, ranging from black-box optimization problems to high-dimensional RL tasks.

Despite reaping the benefits of hardware in order to significantly accelerate the algorithm, there are some limitations that could arise in the future when scaling up this work. As the archive stores the parameters of the entire population among other things, the memory of the device becomes an issue preventing larger networks with more parameters and higher dimensional inputs from being used. This issue showed up at the largest batch size of $131,072$ in the humanoid environment which had a significantly larger observation space of close to 300 dimensions. Similarly, this memory limitation also prevent larger archives with more cells from being used (i.e. larger population size). However, the experiments in this paper do not do use anything less than what is commonly used in QD literature.

QDax is a general framework and tool that is useful for accelerating population-based learning, including QD algorithms, over a wide range of problem settings. From the code provided in the supplementary materials and the experiments shown, we demonstrate that QDax can be used for standard black-box function optimization tasks, simple low-dimensional control tasks and more complex neuroevolution tasks such as in QD-RL. Other than the MAP-Elites (ME) algorithm used in this paper, the QDax framework can be used to implement and accommodate more complex QD algorithms such as ME-ES (Colas et al., 2020), PGA-ME (Nilsson & Cully, 2021), CMA-ME (Fontaine et al., 2020b), Differentiable QD (Fontaine & Nikolaidis, 2021) and more. However, while we expect the findings of the paper in terms of the relationship between the batch size and iterations to hold for these algorithms, the time performance might suffer due to additional computation required such as gradient steps or matrix inversions by these more sophisticated algorithms.

Through this work, we hope the increased accessibility of QDax can help bring ideas from an emerging field of optimization to accelerate progress in machines learning. We also hope to see new algorithmic innovations that will leverage the massive parallelization to improve performance of QD algorithms.

**Broader Impact Statement**

Accessibility was a key consideration and motivation of this work. Our work turns the execution of QD algorithms from what took days/weeks on large CPU clusters to only minutes on a single easily accessible and free cloud GPU/TPU. We hope that this will prevent limitations of access of these algorithms to wider and more diverse communities of people, particularly from emerging and developing economies and beyond only well-resourced research groups and enterprises. We believe this increase in accessibility has the potential to open up novel applications of QD in new domains. On the other hand, a key scientific takeaway from our work was also that QD algorithms were able to scale with more powerful and modern hardware through

massive parallelization. We recognize that this could also open up the possibility of 'buying' results (Schwartz et al., 2020) by utilizing more compute.

We are also aware that algorithms that can scale well with compute come at the risk of carbon footprint cost. Evident from the bloom of deep learning and more recently large language models, computational demands only continue to increase with models that scale. Interestingly, our work shows that given the same machine, we can decrease computation time by two orders of magnitude with no loss in performance of the algorithm which can aid in reducing environmental impact. Nonetheless, we highlight research on the environmental and carbon impact of AI and recommendations to be accountable in minimizing its effects (Dobbe & Whittaker, 2019; Dhar, 2020). A common step towards reducing AI's climate impact is to increase transparency on energy consumption and carbon emissions from computational resources used. We report the estimated emissions used in our experiments in the Appendix. Estimations were conducted using the Machine Learning Impact calculator (Lacoste et al., 2019).

### Acknowledgments

This work was supported by the Engineering and Physical Sciences Research Council (EPSRC) grant EP/V006673/1 project REcoVER, and by Google with GCP credits.

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
