# OpenReview forum: "Accelerated Quality-Diversity through Massive Parallelism"
_TMLR — Accepted by TMLR_

### Review · Reviewer_Lvqf · 2022-11-01

**Summary Of Contributions:**

To speed up runtime of QD algorithms, this paper parallelizes the evaluation of solutions, which is performed in physical simulators when using QD in the field of Reinforcement Learning (RL) for robotics. Unlike previous QD parallel frameworks that use simulators that run on CPUs, this article proposes QDax, which uses simulators that run on GPUs/TPUs, allowing the QD algorithm to take advantage of the hardware acceleration for massive parallelism. The authors also observe through several QD-RL tasks that larger batch sizes and massive parallelism do not negatively impact the final performance of the algorithm when given the fixed number of evaluations, which is interesting.

**Audience:**

Yes

**Broader Impact Concerns:**

The authors have made a comprehensive statement in the Broader Impact Statement section.

**Claims And Evidence:**

Yes

**Requested Changes:**

- I think conducting experiments on more tasks would make the results more convincing, thus strengthens the work.

- The paper misses some recent work in the QD-RL field, such as [1] and [2], adding the reference of which will also strengthen the paper.

[1] Evolutionary Diversity Optimization with Clustering-based Selection for Reinforcement Learning. In: ICLR, 2022.
[2 Promoting Quality and Diversity in Population-based Reinforcement Learning via Hierarchical Trajectory Space Exploration. In: ICRA, 2022.


**Strengths And Weaknesses:**

**Strengths**

- Overall the paper is clear and easy to read and understand. It is well structured and nicely illustrated.
- The experimental evaluation demonstrates the acceleration of the proposed methods and the source code is released, which is good.
- The observation that massive parallelism does not affect the final performance of the algorithm when given the fixed number of evaluations is interesting, which means that linear acceleration is possible within the limit of device.

**Weaknesses**

The contribution of the paper is limited. There have been previous works on parallelism of the QD algorithm which used simulators running on CPUs, while this work applies a simulator from reinforcement learning running on GPUs/TPUs named Brax to the QD algorithm, which was not proposed by the authors but only applied, causing limited contribution.

---

### Review · Reviewer_JCVC · 2022-11-07

**Summary Of Contributions:**

The paper introduces a framework (QDax) for massively parallel evaluations for Quality-Diversity (QD) algorithms and investigates how massively parallel evaluations affect the performance of QD algorithms, both in performance and compute cost. The authors first introduce the broader goal of their method, which is to enable parallel fitness evaluations in QD algorithms leveraging parallel computing in GPUs and TPUs, and introduce related work pertaining to QD algorithms and hardware acceleration. Next, the authors describe the general problem setting of QD optimization and related metrics and outline the MAP Elites algorithm, which serves as the basis of their experiments. Subsequently, the authors describe their implementation of massively parallel implementation of MAP Elites by utilizing JAX based libraries for parallel fitness evaluations of different solutions.

In their experiments, the authors show how large batches of evaluations do not hinder, and in some cases potentially speed up, the convergence of MAP Elites. This is indicated by the smaller of generations required for convergence in the plots for Figure 3. Additionally, the authors how the wall clock time of experiments is significantly reduced using their method (Table 1) and discuss the limitations of their work as well as future work. The results and data presented generally confirm the authors' stated hypothesis.

**Audience:**

Yes

**Claims And Evidence:**

Yes

**Requested Changes:**

1. Add how new solutions $\pi$ are created in Algorithm 1 for greater clarity.
2. Add table summarizing batch size and generations to convergence related to the results in Figure 3. This would help present the data related to less generations required for MAP Elites convergence in a more cohesive way.
3. A discussion on how portable the QDax framework is to other settings (algorithms and optimization problems).

**Strengths And Weaknesses:**

**Strengths**

* The paper is well organized and clearly written with hypothesis outlined and experiments conducted to support the hypothesis.
* The method and problem settings are generally well described and the results are generally well presented.
* The insight provided, such as the one that MAP Elites can converge with less generations on larger batch sizes, is potentially valuable to other researcher in QD algorithms.

**Weaknesses**

* It's unclear how QDax might be used for other algorithms (i.e. not MAP Elites) or other problems settings, such as the ones not presented in this paper. It would have been nice to see a discussion of this in the limitations section.

---

### Review · Reviewer_xwDj · 2022-12-04

**Summary Of Contributions:**

In this paper the authors introduce an study on a subclass of evolutionary algorithms, notably the Quality-Diversity optimizations. The study focuses on hardware acceleration, parallelization and quality related changes due to highly parallel executions. The paper is well written, with appropriate background and previous literature setup. The result of this paper shows an interesting result and that is because of the highly parallel nature of the QD optimization larger parallel problem segmentation can accelerate the solution convergence but without performance improvement.

**Audience:**

Yes

**Broader Impact Concerns:**

There are no any ethical issues that should be of concern. In addition the authors present a proper broader impact section related to the environmental impact due to larger parallelization and thus more power requiring hardware components.

**Claims And Evidence:**

Yes

**Requested Changes:**

Figure 3 is first referenced. Figures should be referenced in order of appearance in the manuscript. Please change this. In addition Figure 1 and 2 are never referenced.



**Strengths And Weaknesses:**

Strengths


The observation of smaller batches and small iterations vs large iterations and smaller batches perhaps indicated that the sampling of even small batches is varied enough to preserve the convergence. In addition the divergence criteria can be strong enough so that the sampling for smaller batches can be still have high enough distinct candidate solutions.

Software scalability

Weakness

I have an issue with the word batch being used here. In general while related to neural networks and in particular when related to neural network the batch represents a set of inputs fed to a network. The main specific point about the batch is that the output or the error form each of the inputs in the batch is then summed and averaged. After which a single error is propagated through the network. So one batch of N samples generates one error E. In the present work this watch confusing to me because the authors refers to batch of solutions but do not average their error or evolution parameters as averaged from all the samples in the batch. While for tracking purpose the QD-score is used it is not used in the algorithm directly. I think a different word should be used here. If this observation is not correct please provide a more detailed explanation of how the batch with a single evaluation is used for updating the solutions in the archive.

The difference between results requires more discussion. As the authors pointed out the results of large batches accelerate certain problems and slow down others. The acceleration of certain problems could mean the search space is small enough or that the landscape of the problem space is well suited for the QD algorithms. But this should be explained and analyzed in details.

---

### Decision · Action_Editors · 2023-01-23

**Recommendation:** Accept as is

**Comment:**

The paper presents, QDax, which leverges massive parallelism on accelerators to speed Quality-Diversity (QD) algorithms, and contributes an open source library for parallelizing QD on a single machine. All reviewers agree, and I concur, that the paper would be a welcome contribution to the TMLR. The authors addressed all the reviewers comments.

**Audience:**

This work is of interest to the Reinforcement Learning subcommunity, as the evaluations and data collections are the major bottlenecks of those methods.

**Claims And Evidence:**

The claims are supported by compelling empirical results.